# "It is because of the love for the job that we are still here": Mental health and psychosocial support among health care workers affected by attacks in the Northwest and Southwest regions of Cameroon

**Moustapha Aliyou Chandini**[1]☯*, **Rafael Van den Bergh**[2]☯, **Agbor Ayuk Agbor Junior**[3], **Farnyu Willliam**[3], **Agnes Mbiaya Mbeng Obi**[3], **Ngu Claudia Ngeha**[3], **Ismaila Karimu**[3], **Emmanuel Christian Epee Douba**[1], **Hyo-Jeong Kim**[2], **Nicholas Tendongfor**[3]

**1** WHO Country Office, Yaoundé, Cameroon, **2** Attacks on Health Care Initiative, WHO, Geneva, Switzerland, **3** Department of Public Health and Hygiene, University of Buea, Buea, Cameroon

☯ These authors contributed equally to this work.
* chandinia@who.int

## Abstract

Attacks on health care have important consequences for the mental health (MH) and work availability of health care workers (HCW). In the conflict-affected Northwest and Southwest (NWSW) regions of Cameroon, health care attacks are common; however, little is known on the MH burden and/or (mental) health-seeking behavior among affected HCW. We therefore conducted a survey on mental conditions (relying on SRQ-20 and WASSS assessments) and access to MH services among 470 HCW from 12 districts in NWSW Cameroon in January-February 2022. In-depth interviews on personal experiences with attacks and on accessing MH services were conducted with a subset of 96 HCW. Among surveyed HCW, 153 (33%) had experienced an attack in the past 6 months, and a further 121 (26%) had experienced attacks more than 6 months ago. HCW facing attacks <6 months ago had significantly higher odds of exhibiting mental disorders (aOR 5.8, 95%CI 3.0–11.3, p<0.001) and of being unable to function (aOR 3.3, 95%CI 1.9–5.7, p<0.001). HCW who experienced an attack >6 months also had higher odds of being unable to function (aOR 2.9, 95%CI 1.7–5.2, p<0.001), and of missing time off work in the week preceding the survey (aOR 3.1, 95%CI 1.8–5.5, p<0.001). Previous access to MH services was also higher among HCW facing attacks. HCW showed a good understanding of the added values of accessing MH services, but faced multiple access barriers (including poor availability of services and their own prioritization of the care of others) and indicated a preference for self-care, peer-support and/or religious support. In conclusion, health care attacks in NWSW Cameroon contributed significantly to severe mental conditions and absenteeism rates among HCW. Strengthening access to MH support among attack-affected HCW is indicated; this should include strengthening of formal MH services and building the capacity of HCW and religious leaders to provide peer-support.

**Data Availability Statement:** As the study reflects data from a high-security/active conflict context, data cannot be made publicly available without potentially endangering study respondents. Anonymised data is available upon reasonable request addressed to the Attacks on Health Care initiative (ahc3p@who.int).

**Funding:** The work was funded by a grant from the Norwegian Ministry of Foreign Affairs to the WHO Attacks on Health Care initiative. The funders had no role in study design, data collection and analysis, decision to publish, or preparation of the manuscript.

**Competing interests:** The authors have declared that no competing interests exist. MAC, RVB, ECED, and H-JK are staff members of the World Health Organization. The authors alone are responsible for the views expressed in this article and they do not necessarily represent the decisions, policy or views of the World Health Organization.

## Introduction

Attacks on health care (here defined as "any act of verbal or physical violence or obstruction or threat of violence that interferes with the availability, access and delivery of curative and/or preventive health services during emergencies" [1]) in fragile contexts around the world represent an important barrier to populations accessing healthcare, and hamper the effort of states to achieve universal health coverage. Attacks on health care include incidents such as high-impact attacks on health facilities, looting and theft of health assets, obstruction of access to health care, and in particular physical and psychological violence towards health care workers (HCW) [2]. The Surveillance System on Attacks on Health Care (SSA) of the World Health Organisation (WHO) documented more than 830 independently confirmed attacks on health care over the course of 2021, more than half of which impacted health personnel [2].

The impact of attacks on HCW has been relatively well-documented in high-income settings, including an emphasis on mental health and psychosocial wellbeing impact and capacity to function [3–6]. Such attacks have been shown to be strongly associated with the development of anxiety, depression, and burnout, leading to high levels of absenteeism [7–9]. Various modalities of providing mental health and psychosocial support (MHPSS) to HCW in the aftermath of critical events have also been investigated in high-income contexts [10, 11]. However, the short- and long-term impacts of health care attacks in low- and middle-income countries, as well as the specific challenges linked to the provision of MHPSS to HCW in such settings, remain poorly understood. Particularly in fragile and conflict-affected settings, where health personnel are already overstretched and overburdened, and where there may be less options for psychosocial support of affected staff, only a small number of studies has been conducted [12–16]. Nevertheless, in such contexts an enhanced understanding of the mental health and psychosocial needs of HCW, and the modalities of offering support that is tailored to the local health-seeking behaviour, is essential, as ensuring access to health care to the population is predicated on the presence and retention of qualified staff.

The Northwest and Southwest (NWSW) regions of Cameroon (hosting over 16% of the country's total population) are examples of such fragile settings, having experienced socio-political instability since October 2016 and seeing grave escalation of insecurity and violence since November 2017 [17]. The region has seen intensive hostilities between the government's armed forces and separatist armed groups, leading to significant internal displacement of more than half a million individuals (as of June 2022), and resulting in a range of attacks against health facilities and HCW [18]. In order to better understand the impact of these attacks on the mental health and psychosocial wellbeing of HCW as well as on their availability for work in this conflict-affected context, and in order to chart the (mental) health-seeking dynamics of HCW in such a context, we conducted a formal study among HCW facing attacks in 12 districts of NWSW Cameroon.

## Methods

### Study design

This was a mixed methods study with an embedded design, in which a small-scaled qualitative component (consisting of a series of in-depth interviews) was integrated in a wider quantitative survey among HCW.

### General setting—Northwest & Southwest regions of Cameroon

The Northwest region Cameroon is situated in its western highlands. The regional capital of the Northwest region is Bamenda, and the population of this region is 1,868,000 inhabitants

according to the 2021 projections of the Ministry of Public Health [19]. The Northwest region is divided into 20 health districts, each headed by a district medical officer. The Southwest region is located along the coastline of Cameroon. The regional capital is Buea and the population is 1,899,900 inhabitants according to the 2021 projections [19]; the region is divided into 19 health districts, each also headed by a district medical officer. The current crisis, resulting from conflict between regional secessionist groups and the central government since 2016, has significantly impacted the health system in the Northwest and Southwest regions with the Health Cluster reporting the closure of up to 21% of health facilities as of February 2022 [20, 21].

## MHPSS services in the Northwest and Southwest regions of Cameroon

Mental health care in NWSW Cameroon prior to the crisis was poorly developed and remained limited to a small number of facilities. The crisis increased the burden on the limited mental health and psychosocial support services as more and more people need mental health care services [22]. Since 2020, WHO and other partners have tried to fill this gap by deploying clinical psychologists and psychiatrists to key health facilities and building the capacities of health personnel and other relevant stake holders to identify and provide psychological first aid (PFA) to victims of trauma [23].

## Study population and study period

Health care facilities in the NWSW Cameroon were purposively selected: first, six health districts were purposively selected in each region, based on their accessibility/security for the study teams. A master list of all health facilities in these districts was compiled, and for each district, health facilities that had reported attacks over the past 5 years and health facilities that had not reported any attacks were selected in a 3:2 ratio. Subsequently, the selected facilities were visited over the course of January-April 2022: facility management was informed of the purpose of the visit, and individual staff members were invited to participate in the quantitative survey. Participants were then categorised based on their self-reported experience of an attack. The selection process is summarised in S1 Fig.

## Sources of data and data collection

The quantitative survey was conducted among all invited staff members of the selected facilities who consented to participate; failure to consent was the only exclusion criteria. Surveys were conducted by trained survey staff, using KoBoCollect v2022.2.3.

Among the participants of the quantitative survey, a purposive sampling was also conducted for participation in the qualitative component. Sampling aimed to recruit participants with diverse backgrounds (general practitioners, nurses, midwives, laboratory technicians and administrative staff), in order to capture the diversity of experiences and perspectives on health seeking behaviour concerning MHPSS care. During the quantitative survey, participants of the qualitative component were asked whether they were willing to participate in an in-depth interview on the topic; if they were willing, an appointment was made at a time and place of their convenience, ensuring maximal privacy for conducting the interview. Interviews were conducted by trained qualitative researchers, in the language of choice of the participants. Interviews were recorded, and transcribed in English (either verbatim for interviews in English or directly translated). Two researchers (AAA and RVdB) conducted a manual thematic analysis, following three pre-defined organizing themes ("personal experience with attacks", "accessing MHPSS support by participants", and "providing MHPSS support to others").

Results were compared, and differences in findings were resolved through iterative discussions.

## Data analysis

While facilities were purposively sampled based on their status as reporting attacks or not reporting attacks, individual participants were categorised based on their personal experiences of attacks (i.e. whether they self-reported experiencing an attack). The self-reported date of the most recent attack they encountered was then used to further categorise respondents as HCW who did not directly experience any attacks, HCW for who the most recent attack occurred up to 6 months prior to the survey, and HCW for who the most recent attack occurred more than 6 months prior to the survey.

Two separate measures were used to assess the participants' mental health conditions: the Self-Reporting Questionnaire-20 (SRQ-20) and the WHO-UNHCR Assessment Schedule of Serious Symptoms in Humanitarian Settings (WASSS) section A [24]. A total SRQ-20 score was calculated for each respondent, which was both represented as is, and was used to identify study participants with common mental disorders: as outlined in [25], men with an SRQ-20 score >8 and women with an SRQ-20 score >10 were considered as having a common mental disorder. For the WASSS, the recommendations in the WHO-UNHCR guideline were followed of considering all individuals reporting a condition "some of the time", "most of the time", and "all of the time" as positive for that condition [24].

To assess which participant characteristics were independently associated with the different outcomes of interest (presence of mental disorders, WASSS, work absenteeism), multivariate logistic regression was performed. In a first step, all characteristics associated with the outcome of interest in a bivariate test (chi$^2$ test, $p<0.1$) were selected. Subsequently, stepwise backward regression was conducted, and only factors significant in the regression ($p<0.05$) were retained. Statistical analysis was conducted using EpiData Analysis v.2.2.3 (EpiData Organisation, Odense, Denmark) and Stata v.17.0 (StataCorp, College Station, TX, USA).

## Ethics

The study was approved by the Institutional Review Board of the Faculty of Health Sciences, University of Buea under application number 1539–12. All participants (both for the quantitative and the qualitative components) provided verbal informed consent, which was witnessed by an independent staff member who was not part of the study. Only the study staff taking the consent had access to identifiable information of the research participants; no other staff had access to information that could identify individual participants during or after data collection.

## Results

### Characteristics of HCW in the Northwest and Southwest regions of Cameroon

A total of 470 HCW participated in the survey: the study population was predominantly female (65%), with representation of medical (53%), paramedical (26%), and non-medical (22%) staff (Table 1). More than half of the respondents had direct experiences with attacks: for 153 (33%), the most recent attack occurred up to 6 months prior to the survey, while 121 (26%) had experienced attacks >6 months ago. Among those facing attacks, the most common types were psychological violence (44%), deliberate obstruction of access (43%) and physical

**Table 1. Characteristics of health care workers participating in a survey on mental health conditions related to attacks on health care in the Northwest and Southwest regions of Cameroon, January-February 2022.**

| Characteristics | N (%) |
| --- | --- |
| Sex: | |
| • Female | 307 (65) |
| • Male | 163 (35) |
| Age: | |
| • 18–29 | 117 (25) |
| • 30–39 | 164 (35) |
| • 40–49 | 120 (26) |
| • 50–59 | 63 (13) |
| • 60+ | 6 (1) |
| Marital status: | |
| • Single | 170 (36) |
| • Co-habiting | 11 (2) |
| • Married | 272 (58) |
| • Divorced | 2 (<1) |
| • Widow(er) | 15 (3) |
| Highest education level: | |
| • None or primary | 41 (9) |
| • Secondary | 77 (16) |
| • High school | 123 (26) |
| • University | 229 (49) |
| Occupation: | |
| • Medical staff (doctor/nurse) | 247 (53) |
| • Medical support staff (pharmacy/lab/imaging) | 123 (26) |
| • Facility staff (sanitary/security) | 45 (10) |
| • Other | 55 (12) |
| Position: | |
| • Director | 27 (6) |
| • Middle-management | 119 (25) |
| • Staff | 259 (55) |
| • Community outreach | 15 (3) |
| • Other | 50 (11) |
| Years of experience | |
| • ≤1 year | 43 (9) |
| • 1–5 years | 134 (29) |
| • 6–10 years | 125 (27) |
| • 11–15 years | 76 (16) |
| • 16–20 years | 47 (10) |
| • >20 years | 45 (10) |
| Experienced attacks | |
| • <6 months ago | 153 (33) |
| • ≥6 months ago | 121 (26) |
| • Never | 196 (42) |

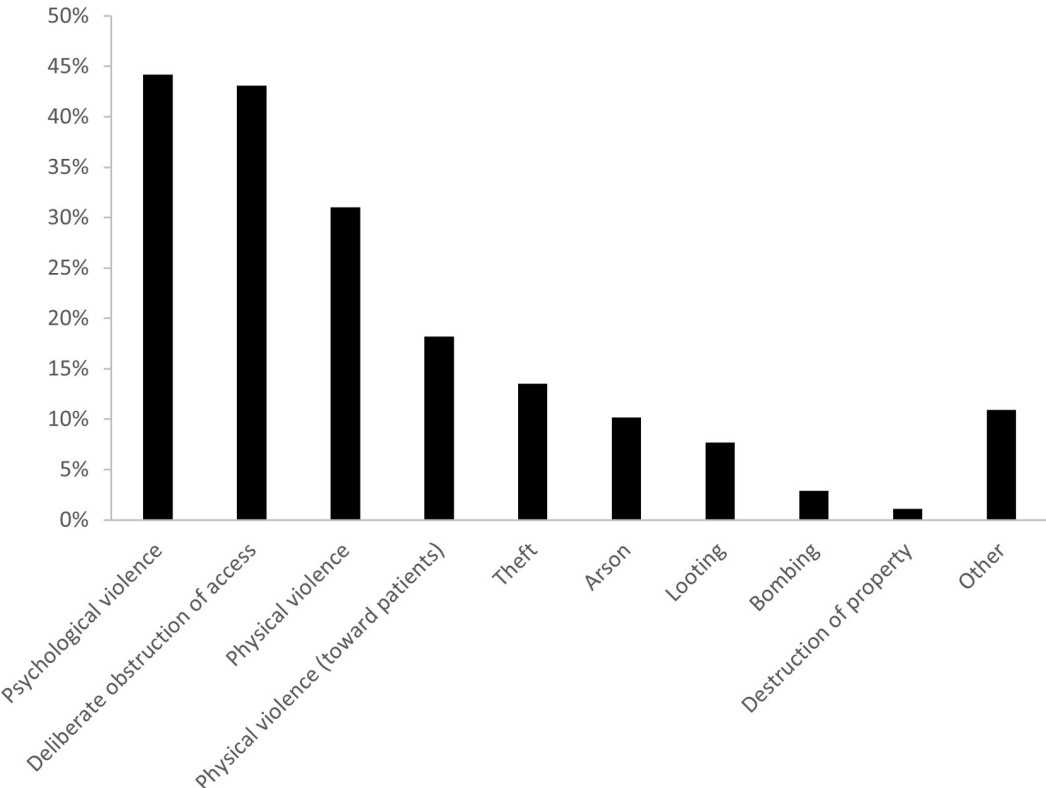

**Fig 1. Types of attacks on health care faced by health care workers in the Northwest and Southwest regions of Cameroon, January-February 2022.**

violence (31%) (Fig 1). During the IDI, HCW echoed these experiences, with most respondents indicating a past experience with at least one and often multiple types of attacks. Threats and extortion of HCW were particularly common:

*One of the worst situations that really made me feel sad is when my mother was kidnapped, because they knew I was a health worker, and they thought that if they get my mum, I have to give them some ransom.*

-Male nurse, age undisclosed

*This attack was from [undisclosed] who blamed us that we were treating [undisclosed] in our hospital. The nurses at the hospital treated (them) too because we were all afraid of the consequences of not treating them. They said they were going to put us in their trucks and that they were going to kill us. They destroyed properties at the hospital buses and some other utilities while searching.*

-Female nurse, age undisclosed

*Those boys have got me well beaten several times. They had beaten me in one occasion until I sustained wounds all over my body and legs in particular. They have been beating me all this time on the count that I work for the government. There was a time when they came and only asked for money, that since I am working for the government, I should pay them money.*

*Recently, they called my wife and threatened us, and this has made it until now, my wife and I are unable to go to the village because of the threats on our lives.*

-Male laboratory technician, age undisclosed

## Attacks on health care are associated with mental health conditions among HCW

Survey participants were grouped in function of their experience with attacks: HCW who did not directly experience any attacks, HCW for who the most recent attack occurred up to 6 months prior to the survey, and HCW for who the most recent attack occurred more than 6 months prior to the survey. Between these groups, the median SRQ-20 scores were plotted (Fig 2A), and the occurrence of common mental disorders (as assessed through the SRQ-20 [25]), the responses to the 5 WASSS questions, and the degree of work absenteeism were compared (Fig 2B). HCW experiencing an attack ≤6 months had significantly higher SRQ-20 scores, and had significantly higher odds of exhibiting mental disorders and of feeling afraid, losing interest, avoiding others, and being unable to function. HCW who experienced an attack >6 months also had higher odds of being unable to function, and of missing time off work in the week preceding the survey (Table 2).

The qualitative findings supported these observations, with most HCW who experienced attacks indicating that these had a considerable impact on their capacity to work, both through direct interruption of service provision and through fear and anxiety among HCW to attend work:

*They took us to the cell. We spend around 36 hours under police detention, for no good reason. So we have been attacked, and that has affected us because the health centre was closed for*

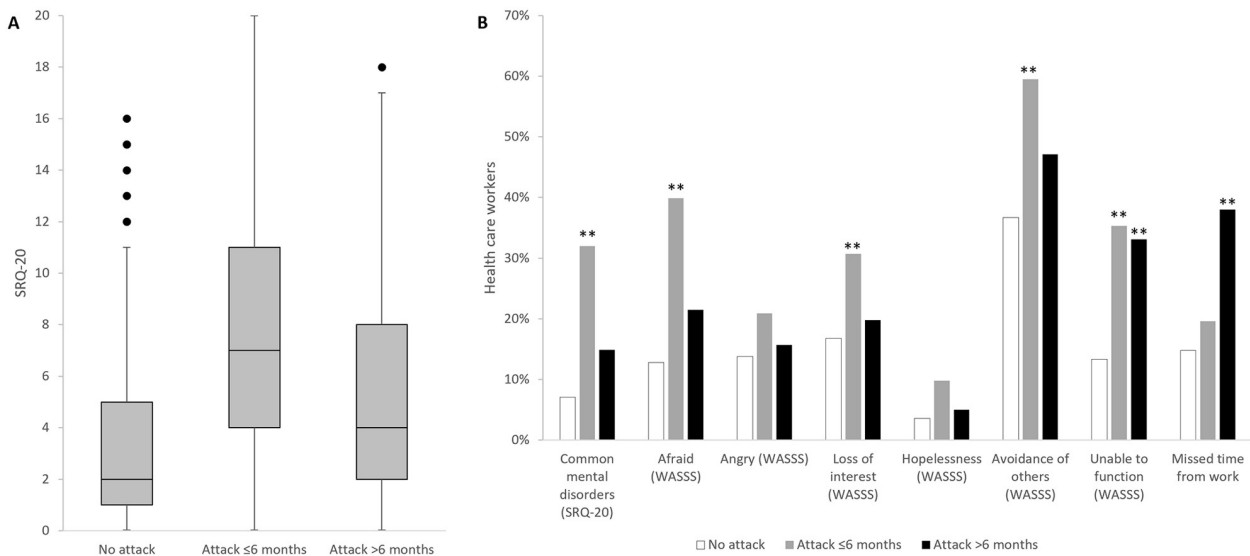

**Fig 2. Tukey box-plot of SRQ-20 scores (A) and proportions of individuals experiencing mental health and psychosocial consequences (B) among health care workers with different experiences of attacks on health care in the Northwest and Southwest regions of Cameroon, January-February 2022.** SRQ: self-reported questionnaire; WASSS: WHO-UNHCR Assessment Schedule of Serious Symptoms in Humanitarian Settings (Individual Interviews).

**Table 2. Mental health and psychosocial conditions among health care workers participating in a survey on mental health conditions related to attacks on health care in the Northwest and Southwest regions of Cameroon, January-February 2022.**

| Condition | Attack ≤6 months | | | Attack >6 months | | |
|---|---|---|---|---|---|---|
| | aOR | 95%CI | p-value | aOR | 95%CI | p-value |
| Common mental disorders (SRQ-20) | 5.8 | 3.0–11.3 | <0.001 | 1.8 | 0.9–3.9 | 0.1 |
| Afraid (WASSS) | 4.2 | 2.4–7.3 | <0.001 | 1.8 | 0.9–3.3 | 0.08 |
| Angry (WASSS) | 1.6 | 0.9–2.8 | 0.1 | 1.2 | 0.6–2.3 | 0.6 |
| Loss of interest (WASSS) | 2.0 | 1.2–3.4 | 0.008 | 1.1 | 0.6–2.0 | 0.7 |
| Hopelessness (WASSS) | 2.6 | 1.0–6.6 | 0.05 | 1.5 | 0.5–4.7 | 0.5 |
| Avoidance of others (WASSS) | 2.2 | 1.4–3.4 | 0.001 | 1.2 | 0.7–2.0 | 0.4 |
| Unable to function (WASSS) | 3.3 | 1.9–5.7 | <0.001 | 2.9 | 1.7–5.2 | <0.001 |
| Missed time from work | 1.2 | 0.7–2.2 | 0.5 | 3.1 | 1.8–5.5 | <0.001 |

aOR: adjusted odds ratio; 95%CI: 95% confidence interval; SRQ: self-reported questionnaire; WASSS: WHO-UNHCR Assessment Schedule of Serious Symptoms in Humanitarian Settings (Individual Interviews)

*about 48 hours because the two main people or consultants were not there and the other key people were not there. About 9 staff were under detention for 36 hours.*

-Male nurse, 34

*On reaching the road with my uniform on, I met with the military officers, who asked me to go back at gun point. This caused a lot of fear and panic to the extent that I was shivering. . . the following day I could not go to work. This did not happen only once.*

-Female nurse, 49

*I have received calls from unidentified people saying "we know you, you work here, and we know your daughter. . . So if you do not cooperate, we will attack you, we will kidnap your daughter in [undisclosed]." All these created fear and panic to an extent that I could not work the following day.*

-Female pharmacy worker, age undisclosed

*Depression is a big part of it that people don't see, people die from watching people die, it is painful watching people die. It is not like malaria that after medication it will go away but this sticks in you and each time you are in a crowd or (with) certain people it plays back completely.*

-Female medical doctor, 39

*I had headache, at times I sit and my heartbeat becomes abnormal without any cause, and when I think of my duty and my work, I am supposed to be on duty, it hurts.*

-Female midwife, age undisclosed

*The truth is, whenever the gunshots start, I become so stressed up. Panic makes me urinate at once and I need to go to stool. As a result of this too, I experience difficulties in sleeping and I do not know if I should start taking sleeping pills. . . and for how long will I take them? In the evening I start watching TV, by 9 pm I'll feel sleepy but I have to force myself to sleep till dawn, which is then not obvious.*

-Male laboratory technician, age undisclosed

## Access to MHPSS services among HCW

Survey participants were queried on their past access to MHPSS services. A total of 91 (19%) of the participants had accessed MHPSS services prior to the survey. Medical staff (aOR 2.4, 95% CI 1.2–4.5, p = 0.009) and HCW experiencing attacks >6 months ago (aOR 2.5, 95%CI 1.4–4.5, p = 0.002) were more likely to have previously accessed MHPSS. While the study was not powered to detect differences in use of services between different groups of HCW, individuals facing attacks seemed less likely to make use of specialised services (psychologists or psychiatrists), and more likely to rely on MHPSS through primary health care providers and/or PFA (Fig 3).

The qualitative interviews further explored the knowledge of participants on MHPSS, and the general practices and experiences with seeking such support in NWSW Cameroon. Overall, the need for and the added value of MHPSS were widely acknowledged by HCW, though often this was referred to in the third person–recognition of the value of MHPSS for patients or other HCW, but not for the interviewees themselves.

*I think what we need here is a professional counsellor, although not for me, but I think my colleagues may need it.*

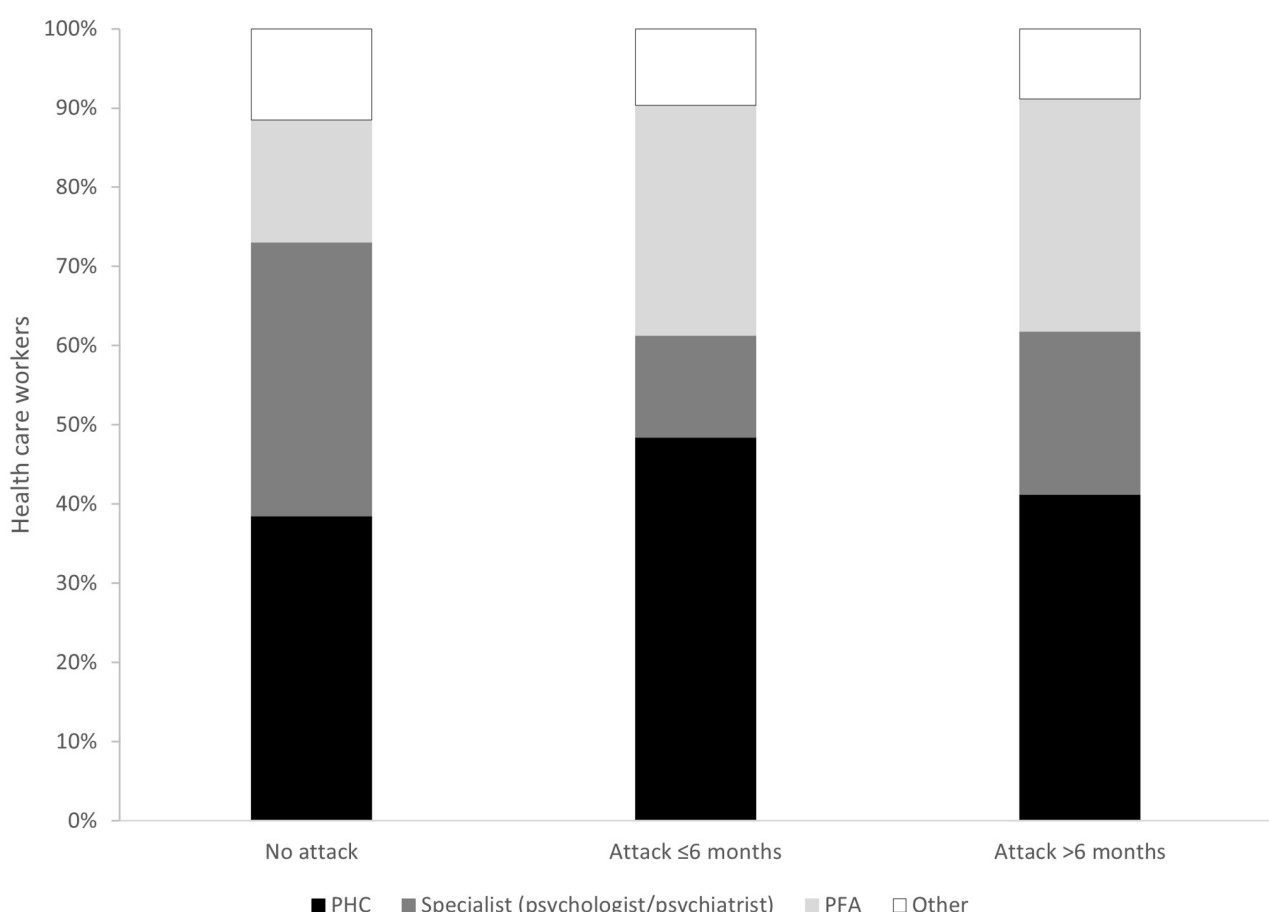

**Fig 3. Types of mental health support options accessed by health care workers with different experiences of attacks on health care in the Northwest and Southwest regions of Cameroon, January-February 2022.** PFA: psychological first aid; PHC: primary health care.

-Female, position and age undisclosed

*I think we need psychological support because in moments when things are bad, for us to leave our homes to come here and vice versa is something very challenging coupled with the fear in our hearts (. . .) (It is) very necessary because we sometimes are devastated with the happenings around us but just a talk with them may help us put ourselves together.*

-Male, position undisclosed, 38

*I think that if I sit and start having some thoughts, like suicidal thoughts, like if I feel that I should die or I should lick poison, I know that I need a psychological counsellor.*

-Female nurse, 52

*Actually I left there satisfied because a counsellor doesn't tell you what to do. It is only advice to try. They don't impose on you what to do. So he advised to do something, which I did.*

-Male nurse, 26

In some instances, however, the value of counselling was questioned, and in some cases was perceived as being too transient and symptom-oriented in a context where HCW face systematic, relentless violence:

*I can say it should not be limited only to counselling. If there is a way to also see how to assist them in one way or the other it will be very necessary because you know that talking to people, and talking over and over, sometimes when you are not assisting them it does not have an impact.*

-Female laboratory technician, 40

*To do what there? Are they free from the fear and the tension? I don't go there at all.*

-Female nurse, age undisclosed

*If I am given psychosocial support and the next day I still face the situation, then nothing will change.*

-Female nurse, 40

Despite the general recognition of the value of formal MHPSS in principle, and of its value for patients in particular, most HCW indicated that they relied on other coping mechanisms in the face of attacks. These commonly included self-care, peer support, deriving strength from caring for others, and support from religious leaders. In some cases, peer support was also seen as a way of referring more severe cases to professional care.

*The first thing is that I try to share with those I think care about me, I go to my pastor and I talk with him, he counsels me and we look at the way forward. At times, I try to get some leisure, just to air myself away.*

-Male nurse, age undisclosed

*I try to share with my colleagues and encourage them, because we all are going through the same thing. But maybe you are stronger than them, or you may have gotten more used to the harassments to the extent that it does no longer affects you.*

-Female, position and age undisclosed

*I talk to them and encourage them, but that could only be done after I had become used to the shock. Also, in some cases, which may seem severe, I recommend them to the sisters for counselling. I also pray with them, and commit everything into Gods hands.*

-Personal details undisclosed

*It is because of the love for the job that we are still here.*

-Male pharmacy worker, 40

Many HCW voiced recommendations on how to improve access to MHPSS services. These included better dissemination of knowledge on the availability of services, sensitisation on the importance of mental health (which many respondents felt was underestimated in African contexts), formalised peer-support networks, and community-level counselling support.

*Where are the counsellors? That is the first thing we should know. You can like to visit a counsellor when you are feeling sad but where will you find them?*

-Male pharmacy worker, 34

*They can also organise campaigns to sensitize the people of [undisclosed] about their mental health and to train the nurses on basic psychological issues, that will be a good way to solve the problem.*

-Male medical doctor, 33

*The truth is that we grew up in an African setting where we don't take mental health problems very serious. That is just the truth, I know that bad things happen but it is left for us not to let the bad things take over.*

-Male medical doctor, 35

*There should be an intentional psychological counselling, be it organized by private organizations or the government, there should be an intentional psychological counselling to all health personnel, not just me who has suffered one form of attack or another, but to all because when you are working in a crisis zone, thinking that the human right law says that a health personnel, no matter the case should not be harmed, we see the reverse.*

-Male nurse, 34

*Mental health workers should reach out to homes. The first thing to do is identify those who are affected so as to easily address their problems.*

-Female nurse, 39

*The first thing is by motivating them. Not only financially. Once you just give a tap on the back of a health worker that "no you are doing great, keep it up", I think the health care worker will keep up. Because some of them may be at gunpoint and asked to do what they are not supposed to do, so by just encouraging them, will be a very good motivating factor to them.*

-Male nurse, age undisclosed

*We need psychosocial assistance, but I also think we need trainings so that we can be self-reliant.*

-Male medical doctor, 31

## Discussion

This mixed methods study assessed the effects of attacks on health care on the mental health of HCW, in a volatile context. Our results indicated that attacks on health care were associated with high levels of mental distress among HCW: workers facing recent attacks had almost six times higher odds of exhibiting common mental disorders, and three times higher odds of feeling unable to function, than HCW who did not face attacks. Even among HCW not facing attacks, mental conditions were not uncommon, reflecting the distress of working in a fragile, conflict-affected environment. Strikingly, mental health consequences (including availability for work) were not limited to the short-term aftermath of the attacks, but persisted more than 6 months after the attack–in particular, being absent from work and feeling unable to function were significantly more common among HCW who experienced attacks >6 months ago, compared to HCW who did not experience attacks, suggesting a dramatic long-term impact of attacks on the availability of the crucial health workforce. Interviewed HCW indicated a good understanding of the added value of MHPSS, though some questioned its relevance in a context of continuous exposure to violence. Despite this understanding of its relevance, most HCW tended to rely on informal and community-based support systems, such as peer support, religious support, and self-care.

While other studies have investigated the link between attacks on health care and psychosocial consequences for HCW in fragile and conflict-affected settings [12–16], to our knowledge this has not been explored in the context of sub-Saharan Africa. Studies in such fragile settings systematically identify the high burden of mental health conditions resulting from violence towards HCW, regardless of the lens through which the study is conducted (burnout, PTSD, anxiety, depression) [12–16]. Similarly, reduced availability for work among HCW experiencing violence is also reported, both in fragile settings and in more stable/high-income environments [15, 26, 27]. Despite the clear impact of attacks on the ability of the health care workforce to continue providing care, relatively few practices for the support of affected HCW are documented. In general, the value of peer support, the availability of reporting mechanisms for incidents of violence, and overall job satisfaction and an encouraging work environment as protection factors for HCW are widely acknowledged [28–30]. There is however a clear dearth of modalities for MHPSS support to HCW in fragile contexts who are exposed to violence, which can be considered as an element of the generalised lack of occupational safety measures in public health in low- and middle-income countries [31].

This study was not without a number of limitations. First, districts in NWSW Cameroon were purposively selected based on their accessibility/security for the study teams: in the conflict reality of this region, random selection was not considered to be acceptable to ensure team security. However, all districts faced similar (high) levels of attacks, and participants from the attacked and non-attacked groups were recruited within these districts with an equal distribution, to be able to control for geographic factors in the analysis. Second, health facilities were purposively sampled to obtain a study population with a mix of exposures to attacks on health care. It is therefore not possible to draw inferences on the prevalence of experience with attacks among HCW. Third, the study was not powered to compare between different types of attacks: follow-up studies assessing the differential impact of e.g. psychological violence versus physical violence would be relevant, to better understand which types of attacks may require more immediate MHPSS. Fourth, the analysis was only based on self-reported data–while standardised self-reporting tools designed for humanitarian contexts were used, no external validation of mental conditions by independent observers was performed. And fifth, study participants were only queried on previous access to MHPSS services in a cross-sectional way, and it was therefore not possible to draw conclusions on the extent to which MHPSS may have

alleviated symptoms of mental distress–here, too, follow-up studies on the performance of MHPSS services among HCW who faced attacks may be indicated.

Despite these limitations, we believe our study carries a number of important operational implications, both in the context of NWSW Cameroon, and in similar contexts facing attacks against health care. The strong associations between attacks and both short- and long-term mental consequences among HCW, including capacity to function and rates of absenteeism, indicate the extent to which attacks may further compromise already weakened health systems by robbing them of their most crucial resource–the health workforce. Health care providers operating in settings where attacks are prevalent are strongly encouraged to systematically implement protection measures, and to develop specific MHPSS services for HCW who have been adversely affected by such attacks.

Such services should take into account the specificities of HCW within a given context. Given the particular reliance of HCW on peer support, religious support, and self-care in NWSW Cameroon, as well as the relatively high usage of PFA among HCW, we suggest that future MHPSS initiatives should utilise these existing processes and make use of the trust that already exists in these support mechanisms. Expanding the cadre of individuals trained on PFA to include religious leaders and community members should be encouraged. This should ideally include training on clear referral criteria for severe mental health conditions, and con-comitant strengthening of specialised MHPSS services to manage referral cases. Additional training should be provided to managers of health care facilities on leadership issues, stress management, and setting up/strengthening peer support programmes in their facilities. In this fashion, existing mental health-seeking behaviour mechanisms and support structures are maintained and strengthened, rather than being supplanted by external services.

Other approaches that should be considered and evaluated in an evidence-based manner include expanding awareness campaigns around MHPSS, particularly focusing on the neces-sity for HCW to address their own needs as well as the needs of those they care for, and pro-active provision of MHPSS to facilities that have faced attacks. As HCW may be reluctant to seek care and/or may not be aware of the nearest services provided, an active outreach to facili-ties that were targeted by attacks and where staff may be in need of emotional debriefing, may prove to be of benefit. Finally, we join our study participants in calling for a rapid cessation of all hostilities, as the only truly sustainable long-term strategy to protect health care.

## Supporting information

**S1 Fig. Study participant selection process for health care workers facing attacks in the Northwest and Southwest regions of Cameroon, January-February 2022.**
(TIF)

**S1 Text. Checklist on inclusivity in global research.**
(DOCX)

## Author Contributions

**Conceptualization:** Moustapha Aliyou Chandini, Rafael Van den Bergh, Emmanuel Christian Epee Douba, Hyo-Jeong Kim, Nicholas Tendongfor.

**Data curation:** Rafael Van den Bergh, Nicholas Tendongfor.

**Formal analysis:** Rafael Van den Bergh, Agbor Ayuk Agbor Junior.

**Funding acquisition:** Moustapha Aliyou Chandini, Hyo-Jeong Kim.

**Investigation:** Moustapha Aliyou Chandini, Agbor Ayuk Agbor Junior, Farnyu Willliam, Agnes Mbiaya Mbeng Obi, Ngu Claudia Ngeha, Ismaila Karimu, Nicholas Tendongfor.

**Methodology:** Moustapha Aliyou Chandini, Rafael Van den Bergh, Agbor Ayuk Agbor Junior, Farnyu Willliam, Agnes Mbiaya Mbeng Obi, Ngu Claudia Ngeha, Ismaila Karimu, Nicholas Tendongfor.

**Project administration:** Moustapha Aliyou Chandini, Rafael Van den Bergh, Nicholas Tendongfor.

**Resources:** Moustapha Aliyou Chandini, Hyo-Jeong Kim, Nicholas Tendongfor.

**Supervision:** Moustapha Aliyou Chandini, Rafael Van den Bergh, Emmanuel Christian Epee Douba, Nicholas Tendongfor.

**Validation:** Moustapha Aliyou Chandini, Rafael Van den Bergh, Agbor Ayuk Agbor Junior, Emmanuel Christian Epee Douba, Hyo-Jeong Kim, Nicholas Tendongfor.

**Visualization:** Rafael Van den Bergh.

**Writing – original draft:** Moustapha Aliyou Chandini, Rafael Van den Bergh, Nicholas Tendongfor.

**Writing – review & editing:** Moustapha Aliyou Chandini, Rafael Van den Bergh, Agbor Ayuk Agbor Junior, Farnyu Willliam, Agnes Mbiaya Mbeng Obi, Ngu Claudia Ngeha, Ismaila Karimu, Emmanuel Christian Epee Douba, Hyo-Jeong Kim.

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
