## [Decision Letter · Decision Letter 0]

21 Jun 2023

PGPH-D-23-00479

“It is because of the love for the job that we are still here”: mental health and psychosocial support among health care workers affected by attacks in the Northwest and Southwest regions of Cameroon

Dear Dr. Van den Bergh,

Thank you for submitting your manuscript to PLOS Global Public Health. After careful consideration, we feel that it has merit but does not fully meet PLOS Global Public Health’s publication criteria as it currently stands. Therefore, we invite you to submit a revised version of the manuscript that addresses the points raised during the review process. Two reviewers have provided constructive feedback, appended below.

We look forward to receiving your revised manuscript.

Kind regards,

Hannah Tappis, DrPH, MPH

Academic Editor

Journal Requirements:

1. Please include a complete copy of PLOS’ questionnaire on inclusivity in global research in your revised manuscript. Our policy for research in this area aims to improve transparency in the reporting of research performed outside of researchers’ own country or community. The policy applies to researchers who have travelled to a different country to conduct research, research with Indigenous populations or their lands, and research on cultural artefacts. The questionnaire can also be requested at the journal’s discretion for any other submissions, even if these conditions are not met.  Please find more information on the policy and a link to download a blank copy of the questionnaire here: https://journals.plos.org/globalpublichealth/s/best-practices-in-research-reporting. Please upload a completed version of your questionnaire as Supporting Information when you resubmit your manuscript.”

2. Our staff editors have determined that your manuscript is likely within the scope of our Global Mental Health: challenges, opportunities, and the future of the field. This editorial initiative is headed by a team of Guest Editors for PLOS GPH: Rochelle Burgess (University College of London) and Dixon Chibanda (University of Zimbabwe and London School of Tropical Medicine and Hygiene). The Collection invites researchers to submit original research which engages with, or disrupts, the urgent needs across the global mental health landscape. We especially encourage submissions of studies that critically interrogate the status quo of the field and that involve inter-/trans-disciplinary approaches and those which share perspectives from underrepresented global regions and communities.

 Additional information can be found on our announcement page: https://collections.plos.org/call-for-papers/global-mental-health-opportunities-challenges/ 

If you would like your manuscript to be considered for this collection, please let us know in your cover letter and we will ensure that your paper is treated as if you were responding to this call.  Please note that being considered for the Collection does not require additional peer review beyond the journal’s standard process and will not delay the publication of your manuscript if it is accepted by PLOS GPH. If you would prefer to remove your manuscript from collection consideration, please specify this in the cover letter.

Additional Editor Comments (if provided):

Reviewers' comments:

Reviewer's Responses to Questions

**Comments to the Author**

1. Does this manuscript meet PLOS Global Public Health’s publication criteria? Is the manuscript technically sound, and do the data support the conclusions? The manuscript must describe methodologically and ethically rigorous research with conclusions that are appropriately drawn based on the data presented.

Reviewer #1: Yes

Reviewer #2: Yes

2. Has the statistical analysis been performed appropriately and rigorously?

Reviewer #1: I don't know

Reviewer #2: Yes

3. Have the authors made all data underlying the findings in their manuscript fully available (please refer to the Data Availability Statement at the start of the manuscript PDF file)?

Reviewer #1: Yes

Reviewer #2: No

4. Is the manuscript presented in an intelligible fashion and written in standard English?

Reviewer #1: Yes

Reviewer #2: Yes

5. Review Comments to the Author

Reviewer #1: I would like to commend the authors for conducting a thorough and insightful study on the impact of attacks on healthcare workers in conflict-affected areas. The study provides valuable insights into the mental health consequences of these attacks, as well as the availability of healthcare workers in such contexts. The authors' use of mixed methods and their comprehensive analysis of the data collected further enhances the robustness of the study's findings. This research is an important contribution to the field and will undoubtedly inform the development of strategies to support healthcare workers in conflict-affected areas

My comments below:

Based on the proposed objective, the study fills a gap in the literature by exploring the impact of attacks on the mental health of HCWs in sub-Saharan Africa and highlighting the need for modalities for MHPSS support for HCWs in fragile contexts exposed to violence.

References:

References need to be included in the following lines: 93 and 96 (population), 103-105, 108. The comparison with other studies in literature in lines 360 and 363 require references as well.

Self-reported data:

The study relies heavily on self-reported data, which can be unreliable due to recall bias or social desirability bias. Participants may have underreported or overreported their experiences or mental health conditions, which could affect the accuracy of the results. The use of objective measures or independent observers could have helped to validate the self-reported data. I believe the Limitation section should be revised to include this limitation.

Secondary trauma:

HCWs who never experienced attacks may have suffered from secondary trauma, which potentially could have affected their ability to provide care as well for fear of being attacked too. It would have provided more insight on the mental health dynamics on the population sampled as a whole rather than just those affected.

Reviewer #2: I would like to point out that I am a French speaker, which could limit my understanding of certain aspects of this article. However, I find the manuscript to be well written overall. I have made a few observations. These are in a Word file that I have attached here.

6. PLOS authors have the option to publish the peer review history of their article (what does this mean?). If published, this will include your full peer review and any attached files.

**Do you want your identity to be public for this peer review?** For information about this choice, including consent withdrawal, please see our Privacy Policy.

Reviewer #1: No

Reviewer #2: **Yes: **Ibrahima BARRY

---

## [Editor Report · Decision Letter 1]

11 Oct 2023

“It is because of the love for the job that we are still here”: mental health and psychosocial support among health care workers affected by attacks in the Northwest and Southwest regions of Cameroon

PGPH-D-23-00479R1

Dear Dr. Van den Bergh,

We are pleased to inform you that your manuscript '“It is because of the love for the job that we are still here”: mental health and psychosocial support among health care workers affected by attacks in the Northwest and Southwest regions of Cameroon' has been provisionally accepted for publication in PLOS Global Public Health.

Best regards,

Hannah Tappis, DrPH, MPH

Academic Editor